# Active Hydraulic Oil Pressure Measurement System as a Source of Information About the Technical Condition of the Aircraft Hydrostatic Drive

**DOI:** 10.3390/s25196031

**Published:** 2025-10-01

**Authors:** Leszek Ułanowicz

**Affiliations:** Air Force Institute of Technology, Księcia Bolesława 6 Street, 01-494 Warsaw, Poland; leszek.ulanowicz@itwl.pl

**Keywords:** aircraft hydrostatic drive, hydraulic system, hydraulic accumulator, corrective element, technical condition, hydraulic precision pair

## Abstract

Current methods for assessing the technical condition of aircraft hydrostatic drives require disassembly of the hydraulic components and their testing on test stands. These methods are expensive and do not provide a quick assessment of their technical condition. The aim of this work is to present the possibility of using an active hydraulic fluid pressure measurement system using a corrective hydraulic accumulator to assess the technical condition of a hydrostatic drive. In the proposed method, the diagnostic information carrier is a change in the corrector’s operation (changes in the course or shape of the output signal), resulting, for example, from wear of the hydraulic pairs of precision hydraulic devices that complete the hydrostatic drive. The output signal from the active pressure measurement system is a control signal that, together with the input interference signal and the output pressure signal, allows for the analysis of changes occurring in the hydrostatic drive. Therefore, based solely on the examination of changes in the corrector’s operation, it is possible to assess various changes occurring in the hydrostatic drive and its hydraulic systems. Studies of pressure step responses in the hydrostatic drive confirmed that hydrostatic drives and their hydraulic systems can be tested using an active hydraulic fluid pressure measurement system with a properly selected corrective hydraulic accumulator.

## 1. Introduction

A high-pressure aircraft hydrostatic drive is a complex technical device consisting of power hydraulic devices connected by hydraulic lines, in which hydraulic oil is one of the structural elements. Hydraulic oil transfers energy from the hydraulic pump (generator) to the hydraulic motor that drives the power receiver. Energy from the hydrostatic drive generator can be transferred to one or more receivers. A characteristic feature of a hydrostatic drive is its closed structure, which involves a circulating hydraulic oil circuit and the ability to vary the unit capacity of its components, enabling smooth control of the power receiver’s output speed. The structure of a hydrostatic drive is illustrated using the example of a typical aircraft hydrostatic drive, the schematic diagram of which is shown in Figure 1.

During the operation of a hydrostatic drive, its technical condition constantly changes due to degressive changes in the hydraulic oil and in the power hydraulic devices that make up the drive [1,2,3]. The dynamic quality of the hydrostatic drive is determined by the current characteristic design dimensions and physical properties of the power hydraulic devices that make up the drive.

Processes occurring in a hydrostatic drive induce dimensional changes, which in turn alter the relative position of the hydraulic device components, observed as increased clearance or frictional instability caused by unbalanced radial fluid pressure forces. Therefore, there is a close relationship between the structural parameters (aging and wear) of a hydraulic device and its operational parameters (efficiency, flow rate, pressure). The technical condition of a hydrostatic drive is therefore closely linked by cause-and-effect relationships with its dynamic state. Wear and aging of a hydrostatic drive are not uniform due to design deviations, manufacturing process differences, and material differences for different components of the same hydraulic device and for the same components within a set of devices of the same type [4,5,6]. Operational practice has shown that research and analysis methods, as well as the resulting technical characteristics, criteria, and principles for sets of hydraulic devices, are not adequate to the properties of individual devices within that set. This is due to the fact that the optimization of the operation process of a set of power hydraulic devices is based on appropriate statistical tests [7,8], while the optimization of the operation process of a single power hydraulic device from a given set of devices is based on appropriate parametric, vibroacoustic, wear, defectoscopic tests, etc. [9,10,11,12].

The operational efficiency of an aircraft hydrostatic drive largely depends on the ability to assess its current technical condition and predict future changes to this condition. Various destructive processes occurring in the power hydraulics of the drive primarily cause flow changes and deterioration of the dynamic performance of the driven object (in aircraft, ailerons, elevators, flaps, landing gear, and air brakes).

Previous methods for assessing the technical condition of aircraft hydrostatic drives are performed periodically and require the removal of the power hydraulics from the aircraft and their examination on test stands. Identifying the technical condition of the hydrostatic drive and its power hydraulics is therefore complex, expensive, time-consuming, and does not provide a quick assessment of its technical condition. It should be noted that hydrostatic drives, including aircraft, are not diagnostically amenable [13,14], meaning that hydrostatic drives lack sensors to measure their basic technical parameters. Therefore, there is a need to improve the diagnostic susceptibility of hydrostatic drives.

According to the author, the current assessment of the technical condition of an aircraft hydrostatic drive can be performed based on changes in inaccessible parameters of its internal structure, determined indirectly based on changes in accessible technical parameters (without disassembling the device). Available technical parameters can include standard measurement signals for aircraft hydrostatic drives, such as pressure, flow rate, and internal leakage flow rate. During tests conducted during aircraft trials, the dynamic state of the hydrostatic drive is measured and recorded. The obtained values and transient waveforms can be compared with reference values and transient waveforms. Therefore, the resulting differences can be transformed into changes in parameters defining the technical condition of the hydrostatic drive.

The aim of this work is to present the possibility of using information from an active system for measuring pressure pulsation or hydraulic fluid flow rate in a hydrostatic drive, using a corrective hydraulic accumulator, to conduct an ongoing assessment of the drive’s technical condition. This measurement involves active and intelligent intervention in the drive structure via a hydraulic accumulator. The corrective element can function in the hydrostatic drive permanently or during the test process. The use of diagnostic information from the active pressure measurement system or hydraulic oil flow rate in the hydrostatic drive is a new approach to the ongoing assessment of the technical condition of the aircraft hydrostatic drive.

## 2. Formulating the Problem and Model of an Active Hydraulic Oil Pressure Measurement System

### 2.1. Corrective Element of the Aircraft Hydrostatic Drive

The dynamic state of a hydrostatic drive can be determined by reducing it to an automatic control system. This is due to the following reasons [15]:-Many input signals (disturbances) acting on hydrostatic drives are sinusoidal in nature (generally, they are the sum of trigonometric functions). This group of disturbances includes, among others, pressure and flow rate pulsations generated by hydraulic pumps. In these cases, the step response characteristics provide direct information about the drive’s behavior.-Dynamic system analysis methods developed in the theory of automatic control are primarily based on frequency characteristics.

The aircraft hydrostatic drive in Figure 1 can be reduced to an automatic regulation system, the block diagram of which is shown in Figure 2. In the resulting automatic regulation system, the control object and the controller can be distinguished, and the automatic system in question can be reduced to a regulation system, the block diagram of which is shown in Figure 3. The setpoint in the block diagram presented in Figure 3 is pressure.

The aircraft hydrostatic drive is, in accordance with the principles of automatic regulation, a regulation system. Therefore, it is possible to purposefully introduce special corrective elements into individual components of this drive, which will improve the static and dynamic properties of the automatic regulation system. The regulation system with the corrective element is shown in Figure 4.

Correction of a hydrostatic drive involves introducing relatively minor modifications to it or its hydraulic systems while maintaining the basic structure of the system and its main components [16]. Correction is achieved by incorporating automation elements called correctors into the hydrostatic drive. The corrector must be selected, in accordance with automation principles, to match the drive’s construction parameters [17]. Correctors are generally introduced into the regulation section of the system, as this is where it is easiest to influence the system’s dynamics. Correctors can be designed for appropriately small signals and low power, which in turn ensures their desirable small dimensions [18,19].

Corrective elements operating permanently in the hydrostatic drive or only for the duration of the test process can be used to continuously assess the technical condition of the hydrostatic drive. Correctors can be used to measure pressure and flow rate pulsations within the drive, providing a source of information necessary for the ongoing assessment of its technical condition. This measurement involves active and intelligent intervention in the drive structure, the so-called active measurement. Active measurement, like measurement with filtering or logarithmization, can be used in diagnostic tests. Any change in the design parameters of the hydrostatic drive resulting from wear of its components will render the originally selected corrector parameters inappropriate. Consequently, this will result in a change in the corrector’s operation, i.e., a change in the waveform or shape of the Q_k_ or p signal (Figure 4) in the hydrostatic drive. Therefore, by examining only the changes in corrector operation, it becomes possible to assess various changes occurring in the drive.

In hydrostatic drives, feedback correction is particularly important due to the need to reduce the impact of disturbances and nonlinear distortions. Feedback is usually negative. It can be easily implemented and is therefore most frequently used [20,21]. Practice confirms that the corrector, by appropriately changing its operation, indicates deterioration of the dynamic quality of the system due to changes in its technical condition [22].

### 2.2. Active Hydraulic Oil Pressure Measurement System in the Hydrostatic Drive

The corrective element, which is the main component of the active sensor for measuring pressure in the drive, is the hydraulic accumulator. A functional diagram of the active system for measuring hydraulic fluid pressure in the drive is shown in Figure 5.

The system shown in Figure 5 consists of the following components:-A measuring tip permanently built into the hydrostatic drive;-A fixed measuring orifice allowing for measuring the effect of the hydraulic accumulator on the hydrostatic drive;-A sensor for measuring pressure difference;-A hydraulic accumulator acting as a corrective element.

The measuring tip allows for connecting the measuring system to the hydrostatic drive line between the hydraulic pump and the actuator.

A fixed orifice allows for the measurement of pressure differences to determine the flow rate of hydraulic fluid supplied to or discharged from the hydraulic accumulator. The required accuracy of flow measurement is ensured by appropriately selecting the technical parameters of the measuring orifice. The operation of a measuring orifice is based on Bernoulli’s principle, which states that two key phenomena occur at a flow restriction: velocity increases while pressure decreases. This relationship allows the measuring orifice to measure the pressure difference on both sides of the restriction. The higher pressure is measured before the restriction, and the lower pressure is measured after the restriction. The pressure difference indicates the flow rate and, therefore, the speed of the hydraulic fluid flow. The measuring orifice consists of a tube mounted in the flow line. Inside, there is a restriction, which increases the velocity at this point. Measuring ports are located on both sides of the restriction, allowing for pressure measurements. The parameters of the measuring orifice, such as: the restriction β of the measuring orifice, the restriction of the orifice use, the dynamic viscosity at the operating temperature, the flow coefficient C, the differential pressure Δp, the mass flow rate Q_m_ or the volume flow rate Q_v_ are determined using the formulas given in the ISO 5167-1:1991 standard [23]. Thus, based on the standard [23], the restriction β of the measuring orifice, i.e., the ratio of the measuring orifice opening d [mm] to the internal diameter of the hydraulic line D (pipe) [mm], should be 0.7. It is also important that the straight pipe section before and after the measuring orifice is sufficiently long. Orifice application limits: 12.5 ≤ d, 50 ≤ D ≤ 1000 (extrapolation), 0.2 ≤ β ≤ 0.75. The orifice allows for a measurement system whose sensitivity is independent of the maximum hydraulic fluid pressure in the hydrostatic drive being diagnosed. To obtain reliable flow rate results, they must be calculated. Bernoulli’s equation is used to calculate the measured pressure difference. When the flow is laminar, accurate results can be obtained without any interference. Turbulence in the flow, less-than-ideal installation conditions, or the presence of contaminants, which negatively impact measurement accuracy, require appropriate corrections to the Bernoulli equation [23]. Therefore, it is always crucial to calibrate the measuring orifice [23], which allows for adapting the results to actual operating conditions, thus obtaining accurate results.

The hydraulic fluid pressure difference sensor allows for measuring its changes at the measuring orifice and, as a result, the flow rate of the hydraulic fluid supplied to the corrector (hydraulic accumulator) or drained from it. Proper operation of the sensor with the desired sensitivity is ensured by appropriately selecting the orifice parameters. The model of the active hydraulic fluid pressure measurement system uses a Siemens SITRANS P Series Z pressure sensor, Type 7MF1564-3DD10-1AA1 (Siemens, Munich, Germany), with a measurement range of 0–250 bar, a maximum measurement error at 25 °C (77 °F), including conformity error, hysteresis, and repeatability of 0.5% of the maximum pressure value, a measurement time of T99 < 0.1 s, and a measurement uncertainty of 0.01 bar.

The hydraulic accumulator is a corrective element. It compensates for leaks, dampens hydraulic shocks, and hydraulic fluid pressure pulsations in the hydrostatic drive. It is a dynamically active element of the drive (regulation system). Its design parameters must be matched to the hydrostatic drive parameters, in accordance with the theory of matching corrector settings to the drive and regulator parameters of the system being tested. The article has been updated with the following text, highlighted in green. The hydraulic accumulator (corrector) is selected based on the required minimum and maximum operating pressure in the hydrostatic drive, the volume of hydraulic fluid to be stored in the accumulator, the initial charge pressure (nitrogen), and the operating temperature. The volume of hydraulic fluid to be stored in the hydraulic accumulator (corrector) should not exceed 0.0015 dm^3^. The required initial pressure on the gas side should be between 60% and 75% of the drive’s maximum operating pressure. A thermodynamic process should be assumed, i.e., slow charging and discharging of the accumulator.

The operation of the active sensor for measuring pressure changes in the aircraft hydrostatic drive is as follows. Pressure pulsation in the hydrostatic drive (Figure 1), resulting from, among other things, wear of the power hydraulic components that complete the drive, will cause the hydraulic accumulator diaphragm (4 in Figure 5) located between the compressed air and the hydraulic fluid to move. This diaphragm movement will cause the hydraulic fluid to flow bilaterally, either into or out of the hydraulic accumulator (4 in Figure 5). A sensor (3 in Figure 5) measures the pressure difference ∆p between the hydrostatic drive and the hydraulic accumulator, and thus the flow rate ±Q_k_. Changes in the pressure difference ∆p (flow rate Q_k_) provide available information that determines the desired effect of the corrector on the hydrostatic drive. Recorded changes in the pressure difference ∆p (flow rate Q_k_), interpreted in close connection with the principles of selecting the corrector settings for the system, can then be linked to the design parameters of the hydrostatic drive.

As previously mentioned, the corrector must be selected, in accordance with the principles of automation, to match the drive’s design parameters [24,25]. We have series, parallel, or feedback correction. In hydrostatic drives, feedback correction is particularly important due to the need to reduce the impact of nonlinear disturbances and distortions present [26,27]. Feedback is usually negative [27]. The general conditions for the correct collector selection result from comparing the value approached at *t*→∞ by the step response *h*(*t*) of the system, before it is subjected to corrective feedback, with the value approached by the step response *h_k_*(*t*) after it is subjected to corrective feedback. Therefore, in hydrostatic drives, for obvious reasons, it is required that:(1)$$\underset{t\to \infty }{\lim}h\left(t\right)=\underset{t\to \infty }{\lim}h_{k}\left(t\right).$$

For drives described by transfer functions, condition (1) takes the form:(2)$$\underset{s\to 0}{\lim}\frac{1}{s}H\left(s\right)=\underset{s\to 0}{\lim}\frac{1}{s}H_{k}\left(s\right),$$
where *H*(*s*) is the transfer function of the system before correction, *H_k_*(*s*) is the transfer function of the system after *H*(*s*) is subjected to corrective feedback.

Condition (2) can be written:(3)$$\underset{s\to 0}{\lim}H\left(s\right)=\underset{s\to 0}{\lim}\frac{H(s)}{1+G_{k}\left(s\right)\cdot H(s)}\;,$$
where *G_k_*(*s*) is the transfer function of the correction element.

Condition (3) will be satisfied when:(4)$$\underset{s\to 0}{\lim}G_{k}(s)\cdot H\left(s\right)=0.$$

Finally, it can be stated that conditions (1) and (4) will definitely be satisfied when *G_k_*(*s*) or *H*(*s*) are differentiating terms.

### 2.3. Diagnostic Model of the Hydraulic System as a Hydrostatic Drive Module of an Aircraft

The aircraft hydrostatic drive shown in Figure 1 can be described by the following equations:(5)$$z-Q_{c}-Q_{ZH}-Q_{o}-Q_{p}=k_{i}\cdot \frac{dp}{dt}\;,$$(6)$$Q_{o}=k_{o}\cdot p,$$(7)$$Q_{ZH}=k_{ZH}\cdot \frac{dp}{dt}\;,$$(8)$$Q_{c}=k_{c}\cdot p,$$(9)$$Q_{p}=\frac{k_{R}}{T_{j}}\cdot \int \left(p_{o}-p\right)\;dt,$$
where *z*—disturbance, e.g., from a change in flow rate resulting from switching on the receiver or from a change in the hydraulic pump capacity resulting from a change in its drive speed;

*Q_c_*—hydraulic fluid flow rate through internal leaks in the hydraulic pump (internal pump cooling);

*Q_ZH_*—hydraulic fluid flow rate downstream of the hydraulic accumulator;

*Q_o_*—hydraulic fluid flow rate to the receiver;

*Q_p_*—hydraulic pump capacity;

*p*, *p_o_*—hydraulic fluid pressure in the hydrostatic drive and set pressure;

*k_i_*, *k_o_*, *k_ZH_*, *k_c_*, *k_R_*—amplification factors;

*T_j_*—time constant.

From the equations describing the aircraft hydrostatic drive shown in Figure 1, the transmittances are determined:(10)$$G_{1}=\frac{p}{z-Q_{c}-Q_{ZH}-Q_{o}-Q_{p}}=\frac{k_{i}}{s}\;,$$(11)$$G_{O}=\frac{Q_{o}}{P}=k_{o},$$(12)$$G_{ZH}=\frac{Q_{ZH}}{P}=k_{ZH}\cdot s,$$(13)$$G_{C}=\frac{Q_{C}}{P}=k_{C},$$(14)$$G_{P}=\frac{Q_{P}}{p_{o}-p}=\frac{k_{R}}{T_{j}\cdot s}\;,$$
where *s* is a complex variable (after the inverse L-1 transformation of time).

In the system shown in Figure 2, we have:(15)$$G=\frac{G_{i}}{1+G_{i}\left(G_{o}+G_{c}+G_{ZH}\right)}\;,$$(16)$$G_{R}=G_{p}.$$

After performing the appropriate calculations, we will get:(17)$$G=\frac{b_{0}}{s+a_{0}}\;,$$(18)$$G_{R}=\frac{b_{1}}{s}\;,$$
where $b_{0}=\frac{k_{i}}{1+k_{i}\cdot k_{ZH}},a_{0}=b_{0}\left(k_{c}+k_{o}\right),b_{1}=\frac{k_{R}}{T_{j}}.$

For the system in Figure 3, we calculate the equivalent transfer function *H*, which is:(19)$$H=\frac{b_{0}\cdot s}{s^{2}+a_{0}\cdot s+a_{1}},$$
where, additionally, $a_{1}=b_{0}\cdot b_{1}.$

The corrective hydraulic accumulator with a fixed measuring orifice (Figure 5) of the active system for measuring changes in hydraulic fluid pressure in an aircraft hydrostatic drive can be described by the equations:(20)$$c_{x}\cdot x=p\cdot A_{p},$$(21)$$x=\frac{1}{\rho \cdot A_{p}}\cdot \int Q_{k}dt,$$(22)$$Q_{k}=R\cdot \left(p_{c}-p_{p}\right),$$
where *c_x_*—stiffness of the diaphragm and compressed gas;

*Q_k_*—mass flow rate of the hydraulic fluid;

*x*—displacement of the hydraulic accumulator diaphragm;

*A_p_*—diaphragm area;

*ρ*—density of the hydraulic fluid; *R*—hydraulic resistance;

*p_c_*—pressure of the hydraulic fluid in the hydraulic accumulator;

*p_p_*—gas pressure in the hydraulic accumulator.

From the equations describing the corrective hydraulic accumulator, its transmittance can be determined:(23)$$G_{k}=\frac{b_{1}\cdot s}{s+a_{2}},$$
where $b_{1}=R,$
$a_{2}=\frac{c_{x}\cdot R}{\rho \cdot A_{p}^{2}}.$

The automatic control system in Figure 2, described by transmittance (19), can be analyzed from the perspective of assessing its sensitivity (the effect of changes in transmittance parameters) to signals in accordance with the principles of automation, including based on the step response. The step responses of the hydrostatic drive pressure were determined using a HEWLETT PACKARD 35/45 system emulator implemented in an EXCEL spreadsheet. The coefficients *a*_0_, *a*_1_, and *b*_0_ are calculated directly from the design parameters (*k_i_*, *k_o_*, *k_ZH_*, *k_c_*, *k_R_*, *T_j_*) of the hydrostatic drive components. The technical data (assumptions) of the corrective hydraulic accumulator are presented in Table 1.

## 3. Research Results and Their Discussion

### 3.1. Simulation Studies of Pressure Changes in a Hydrostatic Drive with Passive Pressure Measurement

Hydrostatic drive diagnostic information can be examined for two system malfunctions: during actuator or hydraulic motor engagement (*z* = ∆*Q_o_*) or during hydraulic pump drive speed increase (decrease) (*z* = ∆*Q_p_*), i.e., a change in pump capacity. A change in the technical condition (design change *a*_0_, *a*_1_, *b*_0_) will result in a specific change in the dynamic state, i.e., a change in the form of, for example, a transient or step response.

A sample six-variant modification, based on changes in the design parameters of a military combat aircraft’s hydrostatic drive, for the case of hydrostatic drive malfunction during actuator or hydraulic motor engagement (*z * = ∆*Q_o_*), is illustrated by the data in Table 2, and for the case of system malfunction during hydraulic pump drive speed increase (decrease) (*z* = ∆*Q_p_*), it is illustrated by the data in Table 3.

Figure 6 and Figure 7 present the step responses for the specified design changes to the hydrostatic drive (object and regulator) listed in Table 3 (variants 1 to 6) and for the actuator activation disturbance (z = ∆Qo). The step response curves show that the specified design change to the drive

-Is not always clearly and unambiguously reflected in a change in the step response curve (e.g., the curves for variant no. 1 in Table 2 in Figure 6 and variant no. 3 in Table 2 in Figure 6);-Is revealed by a change in the step response curve, but with a significant delay (e.g., the curves for variant no. 2 in Table 2 in Figure 6 and variant no. 6 in Table 2 in Figure 7).

Figure 8 and Figure 9 present the step responses for the specified design changes to the hydrostatic drive and regulator listed in Table 3 (variants 1 to 6) and for a disturbance resulting from a change in the rotational speed of the hydraulic pump drive (*z* = ∆*Q_p_*). The step response curves indicate that the change in the nature of the disturbance

-Did not affect the relationship between the design changes and the changes in the step response curves;-Revealed a change in the step response curve with a significant delay (e.g., the curves of variant no. 3 in Table 3 in Figure 8 and variant no. 6 in Table 3 in Figure 9).

Consequently, this leads to situations in which the transformation (which is absolutely necessary in the diagnostic process) of changes in transient step responses to changes in design parameters is very difficult.

Based on the waveforms in Figure 6, Figure 7, Figure 8 and Figure 9, it can be concluded that passive pressure measurement with currently used sensors does not provide clear diagnostic information about the technical condition of the hydrostatic drive. The output signal from the hydraulic accumulator *Q_k_* (see Figure 3) flows out of the hydrostatic drive much faster than the *p* signal. Delays in the *p* signal relative to the disturbance (Figure 6, Figure 7, Figure 8 and Figure 9) can lead to a situation in which information about internal changes in the drive, in the form of a change in the p signal, flows out of the drive distorted after the short-term disturbance has ceased. This prevents effective interpretation of the change in *p*. In this case, the *p* signal becomes of little use in the diagnostic process of hydrostatic drives.

### 3.2. Simulation Tests of an Active System for Measuring Pressure Changes in a Hydrostatic Drive

The control system with the corrective element is shown in Figure 3. Two variants of introducing a corrective hydraulic accumulator into the hydrostatic drive were assumed:-Variant AZ, in which the corrective hydraulic accumulator is introduced into the hydrostatic drive without any structural modifications (see Figure 1 and Table 2 and Table 3);-Variant BZ, in which the corrective hydraulic accumulator is introduced into the hydrostatic drive after disconnecting the hydraulic accumulator (see Figure 1) from the hydrostatic drive (Table 2 and Table 3 with k_ZH_ = 0).

The diagnostic information carrier is the output signal from the *Q_k_* corrector (Figure 3). The sensitivity of the *Q_k_* signal to drive design changes can be determined based on the analysis of step responses for various variants of introducing the corrector into the drive (variants AZ and BZ), various design changes in the drive (Table 2 and Table 3, changes No. 1 to 6), various interference cases (*z* = ∆*Q_o_* or *z* = ∆*Q_p_*), and various correctors (Table 1, assumptions no. 1 and 2). The step responses of the hydrostatic drive pressure for a specific variant of the corrector introduction and interference case were determined using a HEWLETT PACKARD 35/45 system emulator implemented in an EXCEL spreadsheet. The pressure step responses of the hydrostatic drive for variant AZ of introducing a corrective hydraulic accumulator to the hydrostatic drive, the disturbance case *z* = ∆*Q_o_*, design changes in the drive in accordance with Table 2 (No. 1 to 6) and technical data (assumptions) of the corrective hydraulic accumulator in accordance with Table 1 No. 1 are shown in Figure 10 and Figure 11.

The pressure step responses of the hydrostatic drive for variant AZ of introducing a corrective hydraulic accumulator to the hydrostatic drive, the disturbance case *z* = ∆*Q_o_*, design changes in the drive in accordance with Table 2 (No. 1 to 6) and technical data (assumptions) of the corrective hydraulic accumulator in accordance with Table 1 No. 2 are shown in Figure 12 and Figure 13.

The pressure step responses of the hydrostatic drive for variant AZ of introducing a corrective hydraulic accumulator to the hydrostatic drive, the disturbance case *z* = ∆*Q_p_*, design changes in the drive in accordance with Table 3 (No. 1 to 6) and technical data (assumptions) of the corrective hydraulic accumulator in accordance with Table 1 No. 1 are shown in Figure 14 and Figure 15.

The step pressure responses of the hydrostatic drive for variant AZ of introducing a corrective hydraulic accumulator to the hydrostatic drive, the disturbance case *z* = ∆*Q_p_*, design changes in the drive in accordance with Table 3 (No. 1 to 6) and technical data (assumptions) of the corrective hydraulic accumulator in accordance with Table 1 No. 2 are shown in Figure 16 and Figure 17.

The step pressure responses of the hydrostatic drive for variant BZ of introducing a corrective hydraulic accumulator to the hydrostatic drive, the disturbance case *z* = ∆*Q_o_*, design changes in the drive in accordance with Table 2 (No. 1 to 6) at *k_ZH_* = 0 and technical data (assumptions) of the corrective hydraulic accumulator in accordance with Table 1 No. 1 are shown in Figure 18.

The pressure step responses of the hydrostatic drive for variant BZ of introducing a corrective hydraulic accumulator to the hydrostatic drive, the disturbance case *z* = ∆*Q_o_*, design changes in the drive in accordance with Table 2 (No. 1 to 6) at *k_ZH_* = 0 and technical data (assumptions) of the corrective hydraulic accumulator in accordance with Table 1 No. 2 are shown in Figure 19.

The pressure step responses of the hydrostatic drive for variant BZ of introducing a corrective hydraulic accumulator to the hydrostatic drive, the disturbance case *z* = ∆*Q_p_*, design changes in the drive in accordance with Table 3 (No. 1 to 6) and technical data (assumptions) of the corrective hydraulic accumulator in accordance with Table 1 No. 1 are shown in Figure 20. The pressure step responses of the hydrostatic drive for variant BZ of introducing a corrective hydraulic accumulator to the hydrostatic drive, the disturbance case *z* = ∆*Q_p_*, design changes in the drive in accordance with Table 3 (No. 1 to 6) and technical data (assumptions) of the corrective hydraulic accumulator in accordance with Table 1 No. 2 are shown in Figure 21.

The pressure step responses illustrating the operating processes of the hydrostatic drive and corrector indicate the following:(1)Changing the design parameters (k_i_, k_o_, k_ZH_, k_c_) of the hydrostatic drive and regulator components (k_R_, T_j_) causes a change in the step response.(2)The change in the step response as a function of the design parameters (k_i_, k_o_, k_ZH_, k_c_) of the hydrostatic drive and regulator components (k_R_, T_j_) does not depend on the method of disturbing the hydrostatic drive with (∆Q_o_, ∆Q_p_).(3)The change in the step response as a function of the design parameters of the hydrostatic drive and regulator is more pronounced if the hydraulic correction accumulator is the only corrective element operating in the hydrostatic drive, i.e., in variant BZ, when the hydraulic correction accumulator is inserted into the hydrostatic drive after the hydraulic accumulator is disconnected from the hydrostatic drive (see Figure 1).(4)The change in the step response as a function of changes in the design parameters of the hydrostatic drive and regulator depends on the parameters of the corrector used. Therefore, it is possible to use multiple correctors to test the same hydrostatic drive, and the resulting signals, e.g., Q_k1_, Q_k2_, and Q_k3_, significantly facilitate the identification of changes in the hydrostatic drive being tested.(5)During passive measurement of the hydraulic fluid pressure in the hydrostatic drive, diagnostic information contained in the majorant of the step response of signal p (see Figure 6, Figure 7, Figure 8 and Figure 9) is concentrated at a single characteristic point close to the maximum value of the step response of signal Q_k_ (Figure 10, Figure 11, Figure 12, Figure 13, Figure 14, Figure 15, Figure 16, Figure 17, Figure 18, Figure 19, Figure 20 and Figure 21).(6)The slight delay in the step response of the pressure change signal relative to the disturbance allows for effective interpretation of changes in this signal. Information about internal changes in the hydrostatic drive (wear of the hydraulic pairs of precision hydraulic devices) in the form of changes in signal p will flow out of the system undistorted, even after the short-term disturbance ceases.

## 4. Conclusions

To obtain diagnostic information about an aircraft hydrostatic drive, consisting of hydraulic systems that, according to the principles of automation, are control systems, special differential correction elements must be introduced into this drive. The presented active hydraulic oil pressure measurement system, using a corrective hydraulic accumulator, is a system with active and intelligent intervention in the structure of the hydrostatic drive. The corrector of the active hydraulic oil pressure measurement system with transmittance G_k_ must be selected according to the principles of automation (condition (4)) to match the design parameters of the hydrostatic drive and its hydraulic systems. Any change in the design parameters of the hydraulic system with transmittance *H* (resulting, for example, from wear of the hydraulic pairs of precision hydraulic devices that complete the hydrostatic drive) will result in the initially selected corrector parameters with transmittance G_k_ becoming inappropriate, which will consequently be revealed by a change in the corrector’s operation, i.e., a change in the course or shape of the output signal p or Q_k_ in the system. The Q_k_ signal is a control signal that, together with the disturbing input signal *z* and the output signal p, allows for a more in-depth analysis of changes occurring in the hydrostatic drive. Therefore, by examining only changes in the corrector’s operation, it becomes possible to assess various changes occurring in the hydrostatic drive and its hydraulic systems. A design change in the technical condition of the hydrostatic drive (changes in *a*_0_, *a*_1_, *b*_0_) causes a specific change in the dynamic state, i.e., a change in the form of the step response.

Studies of the pressure step responses of the hydrostatic drive with passive pressure measurement (without an active hydraulic oil pressure measurement system) have shown that the output signal Q_k_ from the hydraulic accumulator flows out of the drive much faster than the p signal. Delays in the p signal relative to the disturbance can lead to a situation in which information about internal changes in the drive, in the form of a change in the p signal, flows out of the drive in a distorted manner after the short-term disturbance has ceased. This prevents effective interpretation of the change in p. In this case, the p signal becomes less useful in the diagnostic process of hydrostatic drives.

Incorporating an active hydraulic oil pressure measurement system with a corrective hydraulic accumulator into the hydrostatic drive causes the diagnostic information about the hydrostatic drive, previously contained in the majorant of the step response of the Q_k_ signal, to be concentrated at a single characteristic point close to the maximum value of the step response of the output signal p. In the process of measuring diagnostic parameters, it is easier to analyze the position of a point than a line (a set of multiple points).

Ultimately, it can be concluded that the p or Q_k_ output signals are a diagnostic signal and that hydrostatic drives and their hydraulic systems can be tested using an active hydraulic oil pressure measurement system with a properly selected corrective hydraulic accumulator.

An active hydraulic oil pressure measurement system is a new type of measurement and a source of diagnostic information about aircraft hydrostatic drives. The presented active hydraulic oil pressure measurement system, using a corrective hydraulic accumulator, installed permanently or only for the duration of the diagnostic process of the hydrostatic drive, can be a source of information about the current technical condition of this drive.

## Figures and Tables

**Figure 1 sensors-25-06031-f001:**
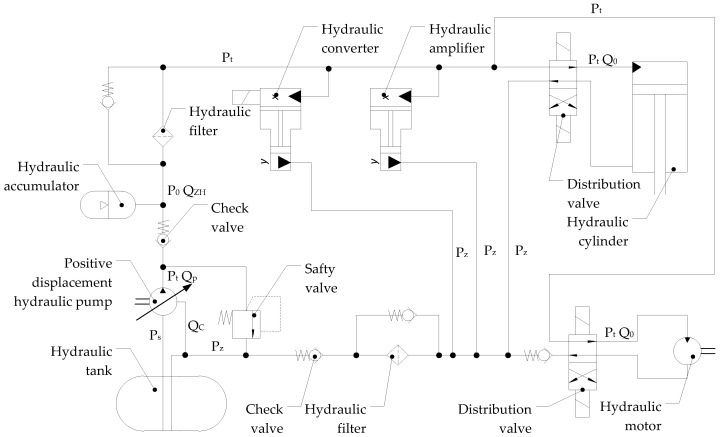
Schematic diagram of a typical aircraft hydrostatic drive.

**Figure 2 sensors-25-06031-f002:**
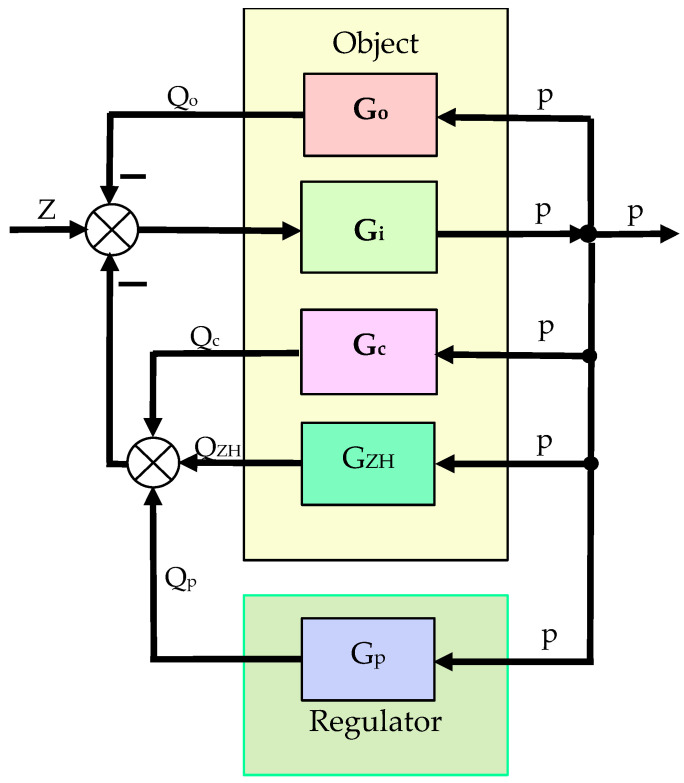
Automatic control system of the considered aircraft hydrostatic drive: Z—disturbance, e.g. from a change in the flow rate as a result of switching on the receiver ∆Q_o_ or from a change in the hydraulic pump capacity resulting from a change in the speed of its drive, Q_c_—flow rate through internal leaks in the hydraulic pump (internal cooling of the pump), Q_ZH_—flow rate from the hydraulic accumulator, Q_o_—flow rate to the receiver, Q_p_—hydraulic pump capacity, p—hydraulic fluid pressure in the hydrostatic drive, G_o_—transmittance of the flow rate to the receiver, G_i_—transmittance of the disturbance, G_c_—transmittance of the flow rate of internal leaks, G_ZH_—transmittance of the flow rate from the hydraulic accumulator, G_p_—transmittance of the hydraulic pump capacity.

**Figure 3 sensors-25-06031-f003:**
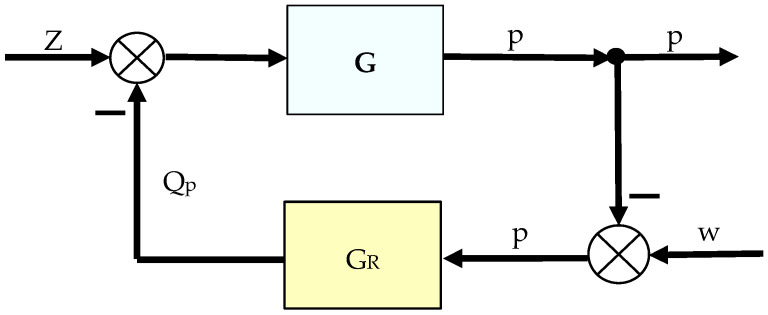
Control system of the considered aircraft hydrostatic drive: z—disturbance (receiver switching on); p—pressure in the drive; w—setpoint value; Q_p_—hydraulic fluid flow rate after the pump; G—equivalent transfer function of the hydrostatic drive (19); G_R_—controller transfer function.

**Figure 4 sensors-25-06031-f004:**
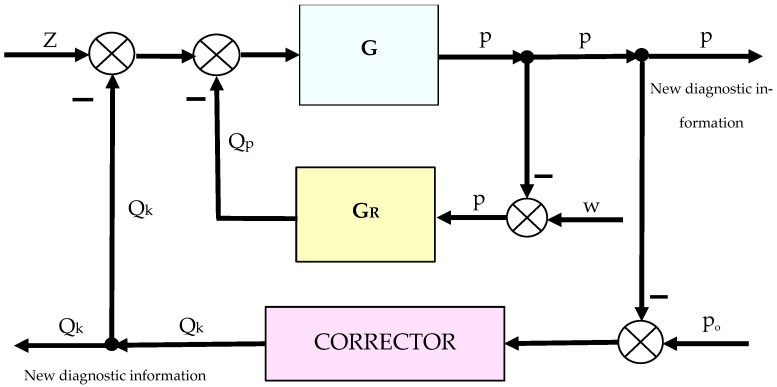
Control system with a corrective element: z—disturbance (receiver switching on); p—pressure in the drive; w—setpoint value to the regulator; p_o_—setpoint value to the corrector; Q_p_—flow rate of the hydraulic fluid downstream of the pump; Q_k_—flow rate from (to) the corrector; G—equivalent transmittance of the hydrostatic drive (19); G_R_—transmittance of the regulator.

**Figure 5 sensors-25-06031-f005:**
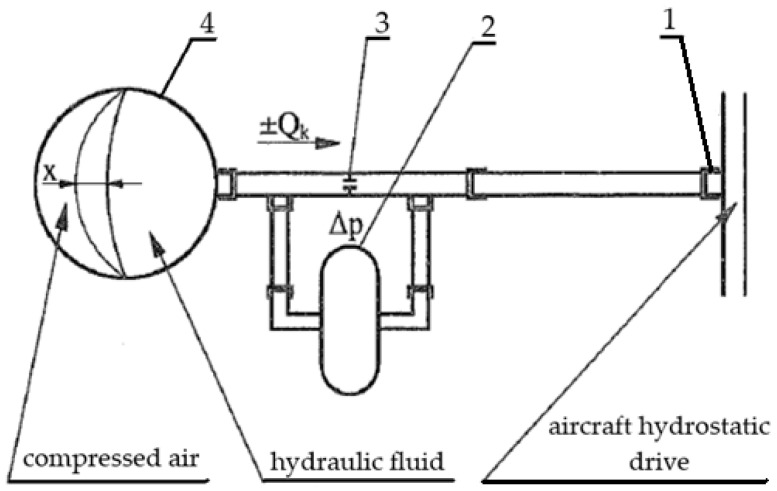
Functional diagram of the system for testing pressure pulsation in the hydrostatic drive of an aircraft: 1—measuring tip; 2—fixed measuring orifice; 3—sensor for measuring pressure difference; 4—hydraulic accumulator acting as a corrective element; ∆p—pressure difference across the orifice; Q_k_—flow rate through the orifice resulting from the corrector’s effect on the drive; x—displacement of the hydraulic accumulator diaphragm.

**Figure 6 sensors-25-06031-f006:**
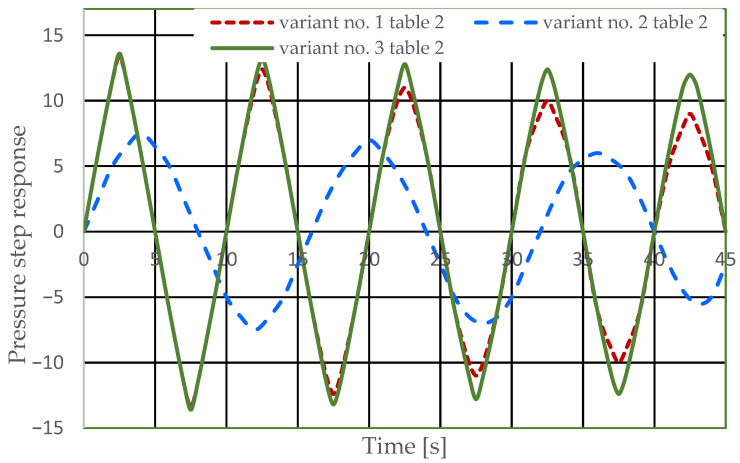
Pressure step response of the hydrostatic drive for variants numbered 1 to 3 in Table 2.

**Figure 7 sensors-25-06031-f007:**
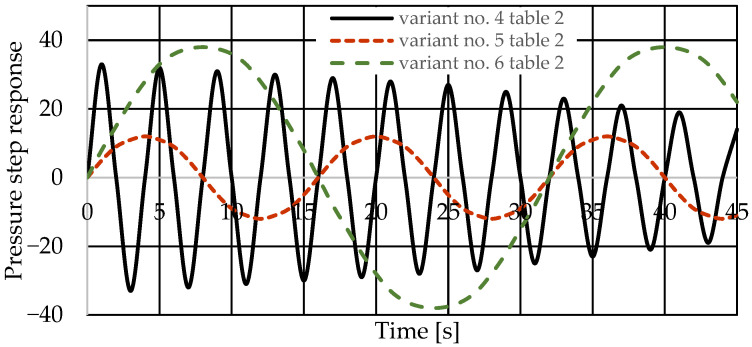
Pressure step response of the hydrostatic drive for variant no. 4 to 6 in Table 2.

**Figure 8 sensors-25-06031-f008:**
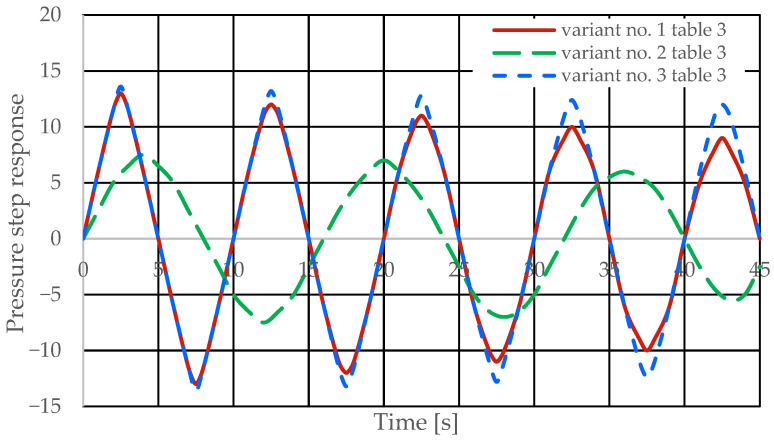
Pressure step response of the hydrostatic drive for variant no. 1 to 3 in Table 3.

**Figure 9 sensors-25-06031-f009:**
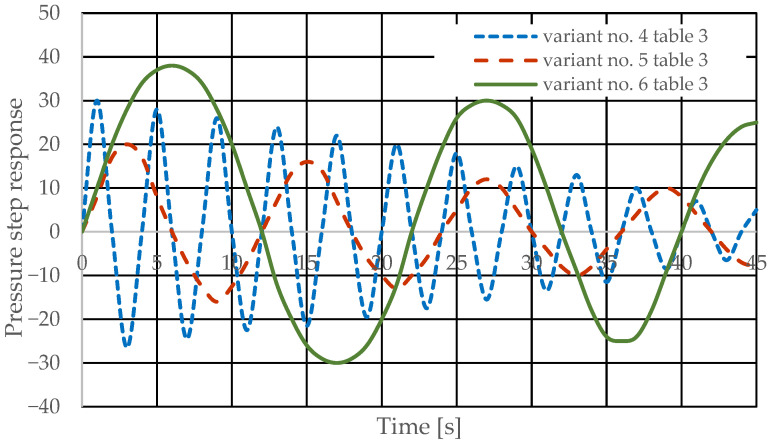
Step response of the hydrostatic drive pressure for variant no. 4 to 6 in Table 3.

**Figure 10 sensors-25-06031-f010:**
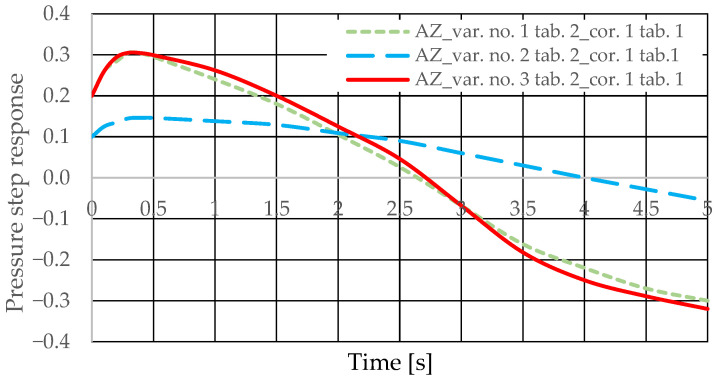
Pressure step response of the hydrostatic drive for variant AZ, cases 1–3 from Table 2, corrector No. 1 Table 1.

**Figure 11 sensors-25-06031-f011:**
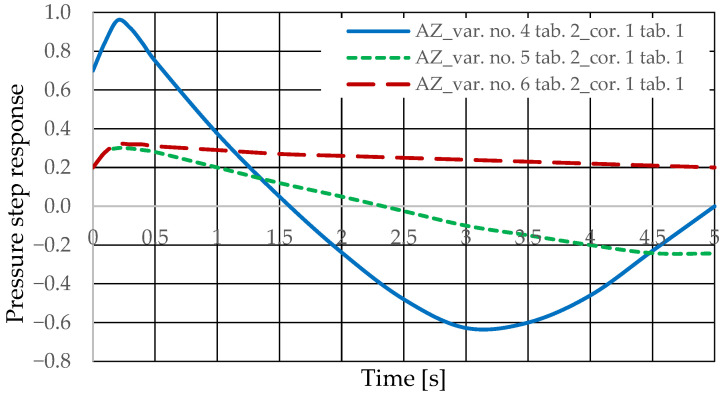
Pressure step response of the hydrostatic drive for variant AZ, cases 4–6 from Table 2, corrector No. 1 Table 1.

**Figure 12 sensors-25-06031-f012:**
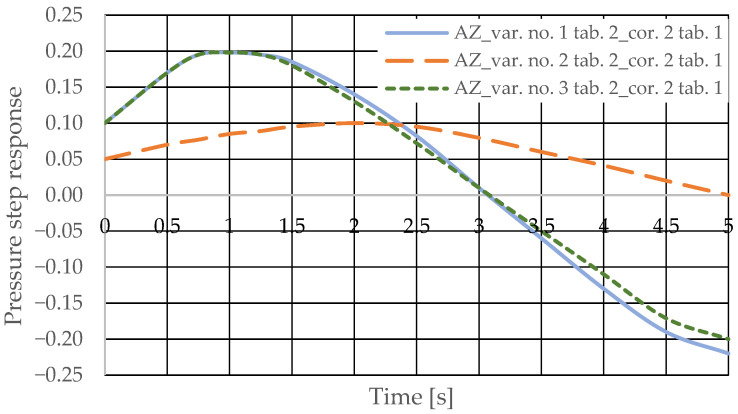
Pressure step response of the hydrostatic drive for variant AZ, cases 1–3 from Table 2, corrector No. 2 Table 1.

**Figure 13 sensors-25-06031-f013:**
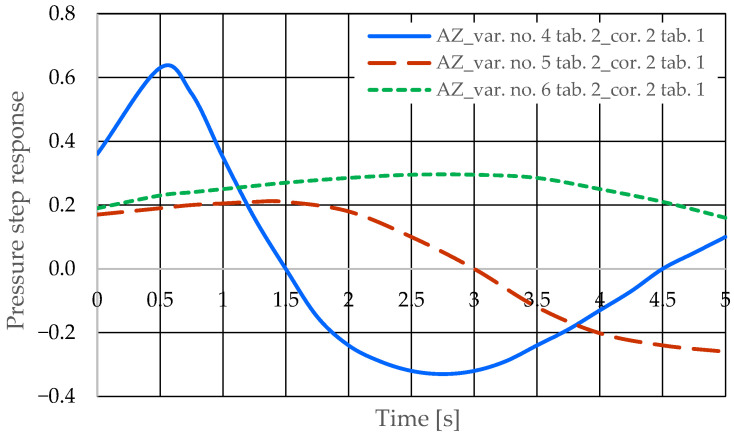
Pressure step response of the hydrostatic drive for variant AZ, cases 4–6 from Table 2, corrector No. 2 Table 1.

**Figure 14 sensors-25-06031-f014:**
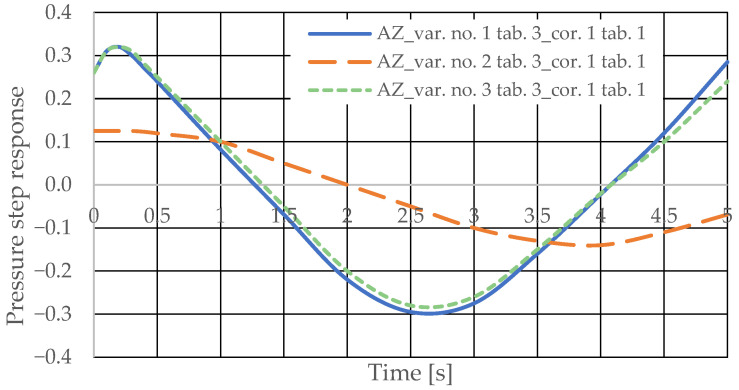
Pressure step response of the hydrostatic drive for variant AZ, cases 1–3 from Table 3, corrector No. 1 Table 1.

**Figure 15 sensors-25-06031-f015:**
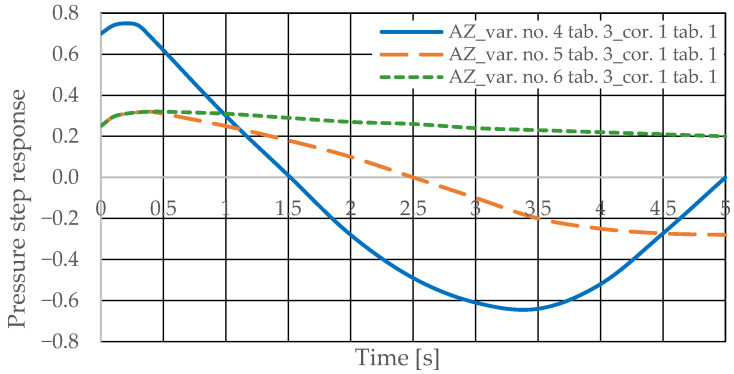
Pressure step response of the hydrostatic drive for variant AZ, cases 4–6 from Table 3, corrector No. 1 Table 1.

**Figure 16 sensors-25-06031-f016:**
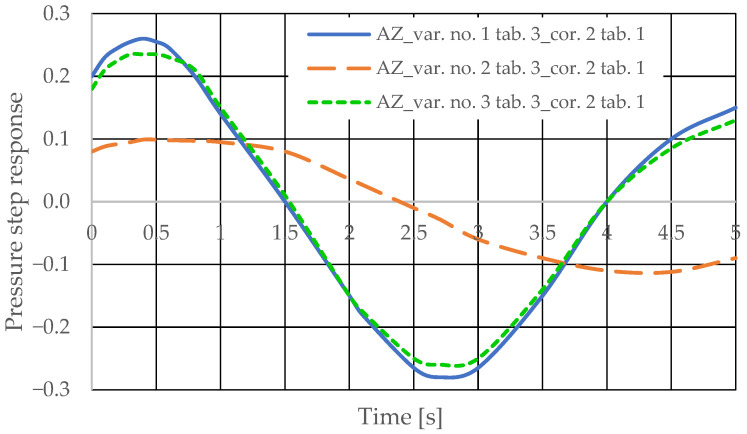
Pressure step response of the hydrostatic drive for variant AZ, cases 1–3 from Table 3, corrector No. 2 Table 1.

**Figure 17 sensors-25-06031-f017:**
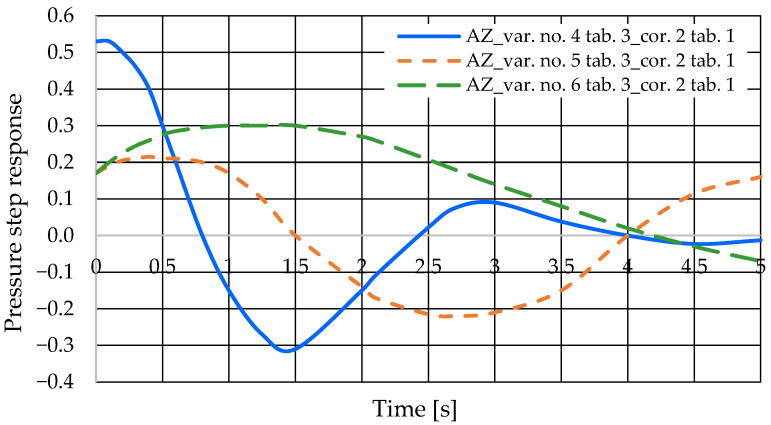
Pressure step response of the hydrostatic drive for variant AZ, cases 4–6 from Table 3, corrector No. 2 Table 1.

**Figure 18 sensors-25-06031-f018:**
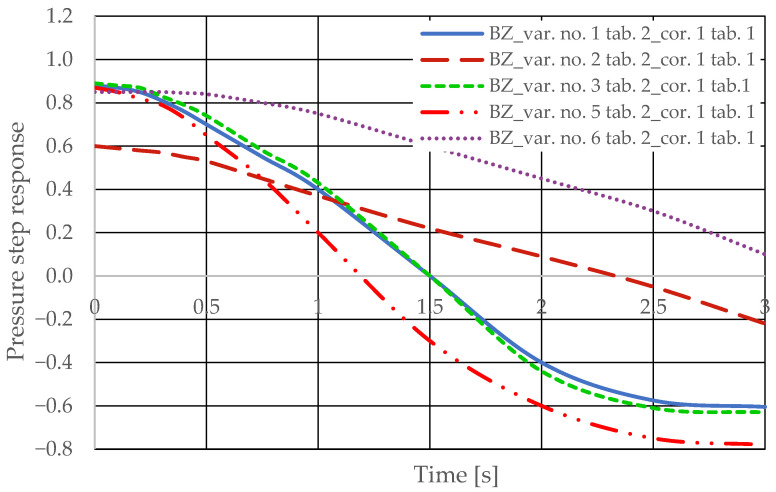
Pressure step response of the hydrostatic drive for variant BZ, cases 1–6 from Table 2 at *k_ZH_* = 0, corrector No. 1, Table 1.

**Figure 19 sensors-25-06031-f019:**
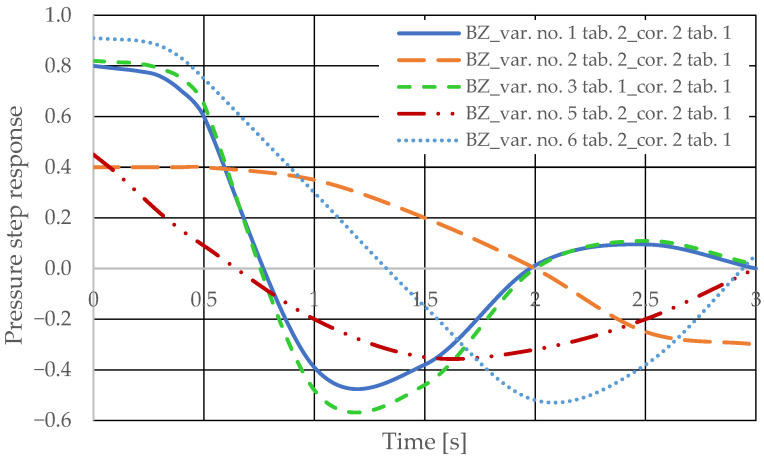
Pressure step response in the hydrostatic drive for variant BZ, cases 1–6 from Table 2 at *k_ZH_* = 0, corrector No. 2 Table 1.

**Figure 20 sensors-25-06031-f020:**
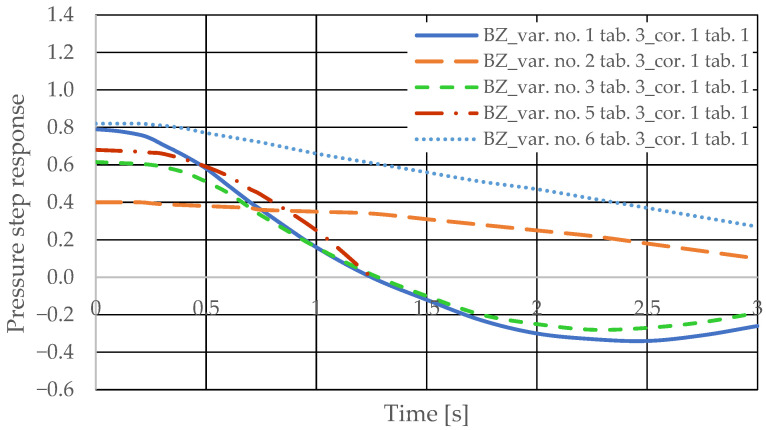
Pressure step response in the hydrostatic drive for variant BZ, cases 1–6 from Table 3 at *k_ZH_* = 0, corrector No. 1, Table 1.

**Figure 21 sensors-25-06031-f021:**
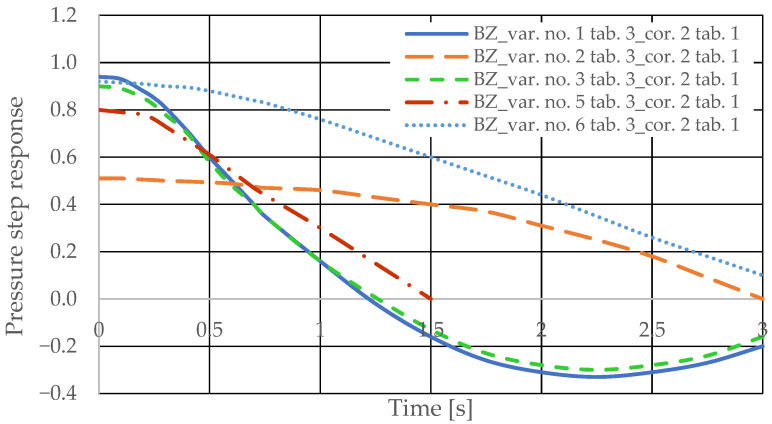
Pressure step response in the hydrostatic drive for variant BZ, cases 1–6 from Table 3 at *k_ZH_* = 0, corrector No. 2 Table 1.

**Table 1 sensors-25-06031-t001:** Technical data of the corrective hydraulic accumulator [daN, cm^2^, cm, s].

No.	*R*	*c_x_*	*b*_1_ = *R*	$a_{2}=\frac{c_{x}\cdot R}{\rho \cdot A_{p}^{2}}$
1	0.4	50	0.4	8.0
2	0.04	50	0.04	0.8

**Table 2 sensors-25-06031-t002:** Technical data of the hydrostatic drive and regulator for the case *z* = ∆*Q_o_* [daN, cm^2^, cm, s].

Lp.	*k_i_*	*k_o_*	*k_ZH_*	*k_c_*	*k_R_*	*T_j_*	$b_{0}=\frac{k_{i}}{1+k_{i}\cdot k_{ZH}}$	$a_{0}=b_{0}\left(k_{c}+k_{o}\right)$	$a_{1}=b_{0}\cdot \frac{k_{R}}{T_{j}}$
1	150	0.003	0.1	0.0002	0.03	0.5	9.4	0.03	0.56
2	15	0.003	0.1	0.0002	0.03	0.5	3.0	0.0096	0.18
3	150	0.0003	0.1	0.0002	0.03	0.5	9.4	0.0047	0.56
4	150	0.003	0.01	0.0002	0.03	0.5	60.0	0.192	3.6
5	150	0.003	0.1	0.00002	0.03	0.5	9.4	0.027	0.56
6	150	0.003	0.1	0.0002	0.003	0.5	9.4	0.03	0.056

**Table 3 sensors-25-06031-t003:** Technical data of the hydrostatic drive and regulator for the case *z* = ∆*Q_p_* [daN, cm^2^, cm, s].

Lp.	*k_i_*	*k_o_*	*k_ZH_*	*k_c_*	*k_R_*	*T_j_*	$b_{0}=\frac{k_{i}}{1+k_{i}\cdot k_{ZH}}$	$a_{0}=b_{0}\left(k_{c}+k_{o}\right)$	$a_{1}=b_{0}\cdot \frac{k_{R}}{T_{j}}$
1	150	0	0.1	0.0002	0.03	0.5	9.4	0.0019	0.56
2	15	0	0.1	0.0002	0.03	0.5	3.0	0.0006	0.18
3	150	0.0003	0.1	0.0002	0.03	0.5	9.4	0.03	0.56
4	150	0	0.01	0.0002	0.03	0.5	60.0	0.012	3.6
5	150	0	0.1	0.00002	0.03	0.5	9.4	0.00019	0.56
6	150	0	0.1	0.0002	0.003	0.5	9.4	0.0019	0.056

## Data Availability

Data confirming the reported results can be found in the Library of the Air Force Institute of Technology in Warsaw. The data generated during the tests are available in the form of reports in the Library of the Air Force Institute of Technology.

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
