# Peer review of "Active Hydraulic Oil Pressure Measurement System as a Source of Information About the Technical Condition of the Aircraft Hydrostatic Drive"

_sensors, 2025, doi:10.3390/s25196031_

Round 1

Reviewer 1 Report

Comments and Suggestions for Authors

This paper proposes an active hydraulic oil pressure measurement system employing a corrective hydraulic accumulator, which can serve as a new source of diagnostic information about aircraft hydrostatic drives. Some suggestions:

1) It is recommended to consider the non-linearity of various components and validate the effectiveness of the system's linear model. In reality, components such as pipelines, orifice throttlers, accumulators, and the squeezing effect of hydraulic oil exhibit obvious nonlinearity.

2) It is suggested to increase experimental research. While ideal systems are easy to measure, actual systems are not; therefore, research based purely on simulation might result in limited engineering value for this paper.

3) Equation (2) contains an error.

4) If the measurement system with corrective hydraulic accumulator is adjusted to be more sensitive, would it affect the dynamic characteristics of the hydrostatic drives?

Author Response

Response to suggestion 1)

I agree with the reviewer that, in reality, the drive components and the hydraulic oil compression effect exhibit obvious nonlinearity. In the current stage of the active hydraulic oil pressure measurement system, I assumed that the power hydraulic devices of the hydrostatic drive were stationary at their operating points with lumped parameters, and I used linear, or more precisely, linearized, mathematical models. In subsequent work, I will consider the nonlinearities of various components and the hydraulic oil compression effect.

Response to suggestion 2)

The material presented in this article primarily concerns a model of an active hydraulic oil pressure measurement system. Laboratory tests of the active hydraulic oil pressure measurement system are planned on a dedicated test stand (currently under construction). The advantage of model-based flow tests, specifically pressure response tests, is the ability to quickly obtain results for a specific operating range spanning several seconds, which is not easy when using other methods and models, such as FEM. I agree with the statement that due to the lack of numerical data from experimental studies and the necessity of relying on the only possible method for their approximate estimation - i.e., curves on graphs - verification of the simulation results was difficult. However, I was able to identify trends or tendencies in the hydrostatic drive's behavior under specific component disturbances.

Response to suggestion 3)

Thank you for pointing out the error. It has been corrected.     

Response to suggestion 4)

Increasing the sensitivity of a measurement system affects the dynamic response of the drive, especially if the sensor begins to detect noise, interference, or respond too quickly to subtle changes, which can lead to dynamic errors and distortions in the measured quantity. The dynamic response of a sensor describes its response to changes in the input signal, and changing this response by increasing sensitivity can impair the system's ability to accurately represent real-world, slowly changing dynamic phenomena.

Reviewer 2 Report

Comments and Suggestions for Authors

Current methods for assessing the technical condition of aircraft hydrostatic drives typically require disassembling hydraulic components and testing them on a bench. These approaches are not only costly but also incapable of enabling rapid condition assessment. To address these limitations, this paper proposes an innovative method that employs an active hydraulic fluid pressure measurement system by incorporating a calibrated hydraulic accumulator. This system enables the online evaluation of the hydrostatic drive’s technical condition and characterizes wear in precision hydraulic components. It allows for comprehensive analysis of the overall status of the hydrostatic drive and its hydraulic system without the need for disassembly. Experimental studies confirm that with a properly selected calibrated hydraulic accumulator, the active pressure measurement system can effectively test the hydrostatic drive and related hydraulic system.

However, there remains considerable room for improvement and refinement. Specific comments and suggestions are as follows:

  1. The parameter matching of the fixed orifice proposed in the paper is critical, as it directly affects the measurement accuracy and range of Qk. Therefore, it is recommended to provide a supplementary explanation of the parameter matching scheme in the article.
  2. The value of flow rate Qk plays a key role in the active pressure control described. It is suggested to include a discussion on the measurement accuracy range and uncertainty analysis of Qk.
  3. The article lacks detailed specifications of the pressure sensor, such as the brand, measurement range, accuracy, and uncertainty. Please provide this information.
  4. The paragraph spanning lines 362 to 404 on page 12 is too long, and the figures are overly concentrated, which hinders readability and understanding. It is recommended to categorize the figures, segment the text accordingly, and align the text with the corresponding figures to improve clarity.
  5. The paper only provides a simulation analysis of the proposed method. It is advised to include experimental validation of key aspects. If conducting aircraft tests is challenging, feasibility and reliability tests should be carried out on a test bench to verify the method’s effectiveness.
  6. The proposed method utilizes an additional hydraulic accumulator to achieve active measurement functionality. However, hydraulic accumulators are generally bulky and heavy, while aircraft hydraulic systems have limited space. Please provide further explanation regarding the practical arrangement of the accumulator to demonstrate the method’s engineering applicability.

Author Response

Response to suggestion 1)

A fixed orifice allows for the measurement of pressure differences to determine the flow rate of hydraulic fluid supplied to or discharged from the hydraulic accumulator. The required accuracy of flow measurement is ensured by appropriately selecting the technical parameters of the measuring orifice. The operation of a measuring orifice is based on Bernoulli's principle, which states that two key phenomena occur at a flow restriction: velocity increases while pressure decreases. This relationship allows the measuring orifice to measure the pressure difference on both sides of the restriction. The higher pressure is measured before the restriction, and the lower pressure is measured after the restriction. The pressure difference indicates the flow rate and, therefore, the speed of the hydraulic fluid flow. The measuring orifice consists of
a tube mounted in the flow line. Inside, there is a restriction, which increases the velocity at this point. Measuring ports are located on both sides of the restriction, allowing for pressure measurements. The parameters of the measuring orifice, such as: the restriction β of the measuring orifice, the restriction of the orifice use, the dynamic viscosity at the operating temperature, the flow coefficient C, the differential pressure Δp, the mass flow rate Qm or the volume flow rate Qv are determined using the formulas given in the ISO 5167-1:1991 standard [23] ("Measurement of fluid flow by means of pressure differential devices, Part 1: Orifice plates, nozzles, and Venturi tubes inserted in circular cross-section conduit - its running full").Thus, based on the standard, the restriction β of the measuring orifice, i.e. the ratio of the measuring orifice opening d [mm] to the internal diameter of the hydraulic line D (pipe) [mm], should be 0.7.It is also important that the straight pipe section before and after the measuring orifice is sufficiently long.Orifice application limits: 12.5 ≤ d, 50 ≤ D ≤ 1000 (extrapolation), 0.2 ≤ β ≤ 0.75.The orifice allows for a measurement system whose sensitivity is independent of the maximum hydraulic fluid pressure in the hydrostatic drive being diagnosed.

The above text has been inserted into the article and highlighted in green.

Response to suggestion 2)

To obtain reliable flow rate results, they must be calculated. Bernoulli's equation is used to calculate the measured pressure difference. When the flow is laminar, accurate results can be obtained without any interference. Turbulence in the flow, less-than-ideal installation conditions, or the presence of contaminants, which negatively impact measurement accuracy, require appropriate corrections to the Bernoulli equation [23]. Therefore, it is always crucial to calibrate the measuring orifice [23], which allows for adapting the results to actual operating conditions, thus obtaining accurate results.

The above text was introduced in the article and is highlighted in green.

Response to suggestion 3)

The article has been updated with the following text, highlighted in green.

The model of the active hydraulic fluid pressure measurement system uses a Siemens SITRANS P Series Z pressure sensor, Type 7MF1564-3DD10-1AA1, with a measurement range of 0 - 250 bar, a maximum measurement error at 25°C (77°F), including conformity error, hysteresis, and repeatability of 0.5% of the maximum pressure value, a measurement time of T99 < 0.1 s, and a measurement uncertainty of 0.01 bar.

Response to suggestion 4)

In accordance with the Reviewer’s recommendation, I modified lines 362 to 404 by categorizing the figures, appropriately segmenting the text, and aligning the text with the relevant figures.

Response to suggestion 5)

The material presented in this article is intended to focus primarily on the model of an active hydraulic oil pressure measurement system and its analysis from the perspective of assessing its sensitivity (the impact of changes in transmittance parameters) to signals in accordance with automation principles and its pressure step response. I agree that due to the lack of numerical data from experimental studies and the necessity of relying on the only possible method for their approximate estimation – i.e., the pressure step response – verification of the simulation results was difficult. However, I managed to identify trends and tendencies in the active pressure measurement system model on pressure step responses to specific disturbances occurring in the hydrostatic drive. To verify the effectiveness of the method, laboratory tests of the active hydraulic oil pressure measurement system are planned on a dedicated test bench. I plan to present the results of the laboratory tests of the active pressure measurement system model in
a future publication.

Response to suggestion 6)

Hydraulic accumulators are used in every hydrostatic drive to compensate for leaks, dampen water hammer, and pressure pulsations in the hydraulic fluid. Instead, an active pressure measurement system can be connected to the hydrostatic drive line port permanently or only for the duration of the test. Unlike conventional hydraulic accumulators used in aircraft hydrostatic drives, the active pressure measurement system can be designed for sufficiently small signals and low power, which in turn ensures the desired small dimensions of the corrective hydraulic accumulator (hydraulic capacity less than 1,5 dm3).

Reviewer 3 Report

Comments and Suggestions for Authors

The paper deals with the development and analysis of an active system for measuring hydraulic oil pressure in aircraft hydrostatic drives, using a corrective hydraulic accumulator. The aim is to enable diagnostics of the technical condition of the drive without disassembly and conventional test-bench procedures, which significantly reduces maintenance costs and time.

The author modestly, but adequately, presents an overview of existing diagnostic methods for hydrostatic drives, including statistical, parametric, and vibroacoustic methods, as well as modern approaches based on artificial intelligence. However, the literature review is more focused on engineering applications and less on a critical comparison of new methods with current trends in diagnostics.

The results are presented through simulations of pressure step responses for different structural parameters of the drive and for different variants of corrective accumulator installation. The author clearly shows that active measurement, unlike passive measurement, enables more precise and faster recognition of changes in technical condition. It is particularly emphasized that the Qk signal from the corrector better reflects internal changes in the drive and facilitates the identification of wear in hydraulic pairs. The analysis is systematic, but it relies exclusively on simulations without experimental confirmation under real operating conditions.

The conclusion logically follows from the simulation results and highlights the advantages of active measurement over passive measurement. However, it lacks:

  • an assessment of the system’s long-term reliability and the potential impact of additional corrective elements on the drive’s operation,
  • a more detailed discussion of the method’s limitations.

Suggestions for improving the paper:

  • Expand the comparison with modern diagnostic methods based on machine learning and multimodal sensors.
  • Define more clearly the criteria for the selection of the corrector and consider the robustness of the system under varying operating conditions.
  • Add a quantitative assessment (e.g., detection error rates or system response times) so that the method can be compared with existing solutions.

Specific points:

  • Harmonize the terminology used in the text of Figure 5 and its caption.
  • In Figure 4, unify the style of symbols (in the figure given as non-italic, in the text given as italic). The same applies throughout the paper.
  • In the diagram of Figure 5, indicate the exact point where the measuring system is connected to the hydrostatic drive line.
  • Clarify the need for Equation (18) to appear at line 152.
  • It is not customary to refer to figures and notations in the conclusion. The conclusion should be reformulated as a textual synthesis of the results, without introducing new figures or formulas, but including both general and specific conclusions derived from the findings.

Overall assessment:
The paper makes a useful contribution to improving diagnostics of aircraft hydrostatic drives, but practical validation and a broader comparative analysis would significantly strengthen its conclusions.

Author Response

Responses to the Reviewer's suggestions for improving the article.

Response to suggestion 1)

Comparison of the active pressure measurement system model with modern diagnostic methods based on machine learning and multimodal sensors will be possible after gaining knowledge based on further research experience, i.e., obtaining so-called training data. The result of knowledge discovery will be changes in the method's parameter values ​​or changes in the method's performance.

Response to suggestion 2)

The article has been updated with the following text, highlighted in green.

The hydraulic accumulator (corrector) is selected based on the required minimum and maximum operating pressure in the hydrostatic drive, the volume of hydraulic fluid to be stored in the accumulator, the initial charge pressure (nitrogen), and the operating temperature. The volume of hydraulic fluid to be stored in the hydraulic accumulator (corrector) should not exceed 0.0015 dm3. The required initial pressure on the gas side should be between 60% and 75% of the drive's maximum operating pressure. A thermodynamic process should be assumed, i.e., slow charging and discharging of the accumulator.

Response to suggestion 3)

Adding quantitative assessments (e.g., detection error rates or system response times) to compare the method with existing solutions will be possible after laboratory testing of the active hydraulic oil pressure measurement system on a dedicated test stand. During the test stand, the active pressure measurement system signal waveforms will be assessed during hydraulic system operation, from the moment the hydraulic pump is turned on and during the extension and retraction of the hydraulic cylinder piston, with simulated hydraulic fluid leakage from the system at specific times.

Responses to specific points made by the Reviewer

Response to suggestion 1)

In accordance with the Reviewer's comment, I have standardized the terminology used in the text of Figure 5 and its caption, i.e., "diagnostic tip" has been replaced by "measuring tip," and "differential pressure sensor" has been replaced by "sensor for measuring differential pressure." I have introduced these changes in the article using green font.

Response to suggestion 2)

Following the reviewer's comment, I have standardized the symbol style in Figure 4 and throughout the article to no longer use italics. I have introduced these changes in the article using green font.

Response to suggestion 3)

In accordance with the Reviewer's comment, in the diagram in Figure 5 I have indicated the "measuring tip" as the point at which the active measuring system is connected to the hydraulic line of the hydrostatic drive.

Response to suggestion 4)

The notation of equation (18) in line 152, i.e. in the caption under Figure 3, was introduced to indicate the formula for the equivalent transfer function described in line 274.

Response to suggestion 5)

In accordance with the Reviewer’s comment, I have reformulated the “Conclusion” as a textual synthesis of the results, without introducing references to figures and formulas.

The material presented in this article is intended to primarily address the model of an active hydraulic oil pressure measurement system and its analysis from the perspective of assessing its sensitivity (the impact of changes in transmittance parameters) to signals in accordance with automation principles and its pressure step response. Despite the lack of numerical data from experimental studies and the necessity of relying on the only possible method for their approximate estimation, i.e., the step response, I managed to identify trends and tendencies of the active pressure measurement system model on pressure step responses under specific disturbances occurring in the hydrostatic drive. To verify the effectiveness of the method, laboratory tests of the active hydraulic oil pressure measurement system are planned on a dedicated test stand.

Round 2

Reviewer 1 Report

Comments and Suggestions for Authors